# The Role of Folic Acid in SARS-CoV-2 Infection: An Intriguing Linkage under Investigation

**DOI:** 10.3390/jpm13030561

**Published:** 2023-03-21

**Authors:** Nikolaos D. Karakousis, Konstantinos I. Gourgoulianis, Ourania S. Kotsiou

**Affiliations:** 1Department of Respiratory Medicine, Faculty of Medicine, University of Thessaly, Biopolis, 41110 Larissa, Greece; 2Faculty of Nursing, University of Thessaly, Gaiopolis, 41500 Larissa, Greece

**Keywords:** COVID-19, folic acid, nutrients, SARS-CoV-2, viral infection

## Abstract

Background: SARS-CoV-2 is a life-threatening RNA virus that may cause an acute respiratory syndrome associated with extremely high morbidity and mortality rates. Folic acid (FA), also known as folate, is an essential vitamin vital for human homeostasis, participating in many biochemical pathways, and its deficiency has been associated with viral infection vulnerability. In this review, we investigated the association between FA intake and SARS-CoV-2 infection, along with the existence of any potential impact of FA on the health outcome of patients suffering from this new viral infection. Methods: Studies included were patients’ and in silico and molecular docking studies. Results: Data from in silico studies and molecular docking support that FA inhibits SARS-CoV-2 entry into the host and viral replication, binding at essential residues. Accordingly, in patients’ studies, a protective role of FA supplementation against SARS-CoV-2 infection is indicated. However, contradictory data from observational studies indicate that FA supplementation, often linked to deficits during systemic inflammation due to SARS-CoV-2, increases the risk of post-infection mortality. Conclusions: Future randomized controlled trial studies, including the FA pharmacological group, are needed to better understand the role of FA as a potential protective or mortality risk indicator in COVID-19 patients.

## 1. Introduction

SARS-CoV-2 is a novel coronavirus from the Coronaviridae family, closely related to the Betacoronavirus virus. Its acute syndrome, attacking the human respiratory system and leading to increased morbidity and mortality rates, was one of the most severe threats to humankind [1]. As of 7 March 2023, there have been 759,408,703 confirmed cases of COVID-19, including 6,866,434 deaths, reported by the World Health Organization [2]. The infection with this RNA virus may have diverse and different outcomes among patients, and targeted interventions are needed [3].

The coronavirus virion consists of the nucleocapsid (N), membrane (M), envelope (E), and spike (S) proteins, which are structural. At the same time, the entry steps of the viral particles are explicitly mediated by the S glycoprotein [4]. Additionally, this type of virus depends on its obligate receptor, angiotensin-converting enzyme 2 (ACE2), to enter cells [5]. SARS-CoV-2 is the seventh coronavirus that is known to infect humans, and along with the Middle East respiratory syndrome (MERS) and severe acute respiratory syndrome (SARS), they are the only ones to cause severe diseases [3]. Many variants of the SARS-CoV-2 virus have emerged and are expected to continue spreading worldwide, giving different symptoms and clinical manifestations [6,7]. Mutations and variants lead to genetic variability associated with an evolving antigen shifting to escape the host immunity [4].

Concerning the therapeutic armory, various drugs are currently being tested. At the same time, vaccines have been developed to build an appropriate preventive wall against this viral infection and protect infected subjects, especially the vulnerable groups of people living among us [3,8,9,10].

It is well established that vitamins might impact different infections, such as tuberculosis (TB), and the use of vitamins is encouraged among subjects with this kind of infection, with vitamins A, B, C, D, and E being in the spotlight [11,12]. In addition, due to its multimodal role in infectious diseases, vitamin D was further investigated for its role. It was demonstrated that this vitamin suppresses, in vitro, the replication of Mycobacterium tuberculosis and appears to have a good profile in managing TB due to its interplay with oxidative balance [13]. 

Data support that vitamin deficiency increases host susceptibility to viral infections [14,15]. More specifically, it has been found that vitamin A supplementation is associated with a better prognosis in the clearance of human papillomavirus (HPV) lesions or reduction in some measles-related complications [14]. The role of vitamin C, selenium, and vitamin D in the immune system against COVID-19 has also been investigated [15]. High doses of Vitamin C have been associated with decreased inflammatory mediators/markers in COVID-19 [15]. Higher selenium levels were linked to a higher recovery rate from COVID-19. Vitamin D deficiency/insufficiency observed in patients with COVID-19 has been associated with a worse prognosis [15].

Folic acid (FA) is a well known small molecule, named vitamin B9, that is related to many biochemical pathways [16]. It is also well established that this vitamin is important in securing health due to its essential participation in the biosynthesis of amino acids, neurotransmitters, nucleotides, and specific vitamins [17]. 

Folate is an important nutrient essential for human homeostasis. At the same time, it has a pterine core structure and high metabolic activity due to its ability to accept electrons and react with S-, O-, N-, and C-bounds [18]. Folate is a pivotal cofactor in essential one-carbon pathways donating methyl groups to creatine, epinephrine, choline phospholipids, and DNA [15,16]. Other compounds with similar features to folate are ubiquitous and have been found in different plants, animals, and other microorganisms [18]. 

It is well established that mammals cannot synthesize folate. Consequently, the primary sources of this nutrient are FA-fortified foods and FA-containing dietary supplements. However, FA is inactive in the human body, and consequently, it must be converted into the active molecule 5-methyltetrahydrofolate (5-MTHF) by the human liver [19,20]. 5-MTHF is associated with many metabolic reactions and functions as a methyl donor. Among these reactions is the biosynthesis of glycine from serine, the conversion of homocysteine into methionine, and DNA precursor molecule biosynthesis [21]. The normal range in the blood is 2.7 to 17.0 nanograms per milliliter (ng/mL) or 6.12 to 38.52 nanomoles per liter (nmol/L) [20]. The values below 2.5–3 lg/L (6–7 nM) are associated with folate-responsive megaloblastic anemia. It has been shown that FA deficiency also decreases intracellular folate [20]. Deficiencies concerning folate might be associated with hereditary folate malabsorption, cerebral folate deficiency, poor diet, obesity, alcohol consumption, and kidney failure [16].

Folate seems to have a protective role and impact on neural tube defects, neurological diseases, and cancer. In contrast, a folate deficiency leads to adverse health outcomes, including anemia, impairments in reproductive health and fetal development, and many others [18,20,21,22,23,24,25,26,27,28]. 

In this review, we aimed to investigate the potential interplay between FA levels and SARS-CoV-2 and demonstrate any potential impact of vitamin intake on the health outcome concerning this viral infection.

## 2. Materials and Methods

We conducted an electronic search in the databases of PubMed, EMBASE, and Google Scholar from 1 March 2020 until 22 December 2022 using the following combinations of specific keywords: SARS-CoV-2” OR “COVID-19” AND “folic acid” OR “folate” OR “vitamin B9”. Only original articles written in English were included in our non-systematic review article. Moreover, the references of included studies were thoroughly examined. Studies concerning animals were excluded from this study. Figure 1 shows the flowchart of the study. 

## 3. Results

### 3.1. In Silico Studies and Molecular Docking Concerning Folic Acid and SARS-CoV-2 Potential Linkage

Table 1 presents the in silico and molecular docking studies concerning the potential association between FA and SARS-CoV-2. The study by Kaur et al. recorded that FA is found to bind to furin-protease and the spike protein–human ACE2 interface of SARS-CoV-2 [29], indicating an interplay between this nutrient and the infection that should be further investigated [29].

In the study by Chen et al., molecular docking showed that FA could act on SARS-CoV-2 nucleocapsid phosphoprotein (SARS-CoV-2 N) [30]. In the same study, 8355 drug targets were used in inhibiting SARS-CoV-2, while 113 hub genes were screened by further association analysis between targets and virus-related genes [30]. The hub genes-related compounds were also analyzed, and FA was screened as a potential new drug [30]. Molecular docking showed that FA could target SARS-CoV-2 N, which inhibits host RNA interference (RNAi). At the same time, FA antagonizes the regulatory effect of SARS-CoV-2 N on host RNAi, concluding that FA might be an antagonist of SARS-CoV-2 N [30]. However, its impact on viruses should be further investigated, as it remains unclear [30]. 

Ugurel et al. analyzed 3458 SARS-CoV-2 genome sequences, which were isolated worldwide [31]. Concerning mutations, the incidence of C17747T and A17858G mutations was much higher than others, and it was also observed to be higher as it relates to Nsp13, a vital enzyme of SARS-CoV-2 [31]. In silico methods were used to estimate the effect of these mutations on protein–drug interactions. The most potent drugs were identified to interact with the key and neighbor residues of the active site responsible for ATP hydrolysis [31]. FA was among the most potent drugs inhibiting wild-type and mutant SARS-CoV-2 helicase [31]. 

Serseg et al. investigated the inhibitory effect of some natural compounds against the 3CL hydrolase enzyme using the molecular docking approach, since the inhibition of the 3CL hydrolase enzyme is supported to be a promising therapeutic principle for developing treatments against COVID-19 clinical syndrome. At the same time, the 3-chymotrypsin-like cysteine protease (3CLpro, Mpro) is known for its involvement in counteracting the host’s innate immune response [32]. A molecular docking study was carried out using Autodock Vina and identified three candidate agents that could inhibit the main protease of coronavirus. It was demonstrated that hispidin, lepidine E, and FA are bound tightly in the enzyme. As a result, strong hydrogen bonds have been formed (1.69–1.80 Å) with the active site residues [30], leading possibly to a potential therapeutic strategy using these three candidate agents that inhibit the main protease of this virus [32].

In another study by Kumar et al., several nutraceutical compounds against known therapeutic targets of SARS-CoV-2 using molecular docking were examined [33]. The structure of all the SARS-CoV-2 targets was retrieved from the RCSB protein data bank. In contrast, a total of 106 compounds from this library (the latest release (2020-1) of nutraceuticals) was retrieved from the drug bank database in Simulation Description Format (SDF) format [33]. The virtual screening results were analyzed for interactive residues and binding energy, and they were additionally compared with some already known hits in the best binding pose. The analyses that were carried out demonstrated that FA alone or in combination with its derivates, such as tetrahydrofolic acid and 5-methyl tetrahydrofolic acid, could be potential molecules against SARS-CoV-2 and COVID-19 infection [33]. This study supported that folates had binding energies similar to or better than those for known drugs targeting these SARS-CoV-2 proteins [33].

Eskandari tried to identify repurposing drugs through in silico screening, docking, and molecular dynamics simulation, since the viral 3CLpro enzyme is vital and quite important for the viral life cycle and controls coronavirus replication, along with the concept that SARS-CoV-2 enters the cell by interacting with the human ACE2 receptor through the receptor-binding domain (RBD) of spike protein [34]. He identified FA, bentiamine, benfotiamine, and vitamin B12 against the RBD of S protein, as well as FA, bentiamine, fursultiamine, and riboflavin against 3CLpro [34]. In this context, the binding of FA at the important residues (R403, K417, Y449, Y453, N501, and Y505) in the S-protein–ACE2 interface and 3CLpro binding site residues, and especially active site residues (His 41 and Cys 145), indicate that it could be a repurpose drug for inhibiting SARS-CoV-2 entry into the host and against viral replication [34].

### 3.2. Patients’ Studies Investigating the Interplay between SARS-CoV-2 and Folic Acid Administration

It is important to seek scientifically proven answers concerning FA’s impact on real-world cohorts of infected patients. Table 2 presents the patients’ studies concerning the potential interplay between FA and SARS-CoV-2 viral infection.

Acosta-Elias et al. investigated the folate concentration and/or FA metabolites in plasma as a risk factor for COVID-19 infection vulnerability [35]. According to the California Department of Public Health, pregnant women were 9.6-fold more likely to be hospitalized during the 2009 influenza outbreak when compared to non-pregnant women of reproductive age, while it was recorded that, among 16,749 COVID-19 patients who were hospitalized in the UK, the probability for pregnant women to require inpatient care due to infection by SARS-CoV-2 was only 0.95 versus non-pregnant women. That concludes that pregnant women are 10.10-fold less likely to be hospitalized for a SARS-CoV-2 infection than for the 2009 H1N1 pandemic [35]. A limitation of this paper was that they had not measured FA in blood. It was also interesting that pregnant black women in the same UK study were 8-fold more likely to require inpatient care than white pregnant women counterparts. It was pointed out that genetic differences may account for differences in folate metabolism, absorption, and red blood cell concentrations. These data indicate that any clinical trial conducted would have to account for differences in race/ethnicity [35]. It would be predicted from the UK study that pregnant black women may require higher dosing of folate to achieve the same level of protection from SARS-CoV-2, or that availability and cost of folate supplementation may prevent use in pregnant women of color in underserved and marginalized communities. Moreover, lower hospitalization rates and inpatient care in pregnant versus non-pregnant women must be due to some other factor of pregnancy, perhaps hormonal, which is unidentified.

Conversely, as stated by Topeless et al., folate supplementation may aid the production of large amounts of the virus through a simple biosynthetic role, thus calling into question all the in silico findings mentioned above [36]. Topless et al. investigated FA and methotrexate usage and their association with COVID-19 diagnosis and mortality using the UK Biobank [36]. There were specific criteria for COVID-19 diagnosis, such as 1. a positive SARS-CoV-2 test or 2. an ICD-10 code for confirmed COVID-19 (U07.1) or probable COVID-19 (U07.2) in the hospital or death records. According to these criteria, 26,003 individuals were identified with COVID-19, of whom 820 were known to have died from SARS-CoV-2 infection [36].

Further statistical analysis was carried out using logistic regression statistical models, which were adjusted for age, ethnicity, sex, Townsend deprivation index, body mass index (BMI), presence of rheumatoid arthritis, smoking status, sickle cell disease, and use of other medications, such as anticonvulsants, statins, and iron supplements [36]. Compared with subjects prescribed neither FA nor methotrexate, subjects prescribed FA supplementation had an increased risk of a diagnosis of COVID-19 (OR: 1.51, 1.42–1.61) [36]. Subjects prescribed FA supplementation had a positive association with death after COVID-19 diagnosis (OR: 2.64, 2.15–3.24) in a fully adjusted model, while the methotrexate prescription in combination with FA was not associated with an increased hazard for COVID-19-related death (OR: 1.07, 0.57–1.98) [36]. This observational study suggested an association between a high risk for COVID-19 diagnosis and COVID-19-related death in subjects treated with FA supplementation, while an antifolate, such as methotrexate, diminished these associations [36].

In a retrospective, observational study by Bliek-Bueno et al., medications associated with increased mortality risk in SARS-CoV-2 infected subjects were evaluated. The study included 8570 individuals from two regions in Spain and Italy using real-world data [37]. They followed the sample for a minimum of 30 days to allow sufficient time for the studied event, in this case, death. The variables concerning the baseline demographic and all drugs dispensed in community pharmacies approximately three months prior to SARS-CoV-2 infection were extracted from the PRECOVID Study cohort (Aragon, Spain) and the Campania Region Database (Campania, Italy), and their analysis was held by using logistic regression models [37]. FA was significantly associated with a 30-day mortality in subjects with COVID-19 in both regions (OR: Aragon, OR: Campania), (OR: 1.59, OR: 2.70) [37]. Other drugs associated with increased mortality from both countries were potassium-sparing agents, vasodilators, high-ceiling diuretics, antipsychotics, antithrombotic agents, vitamin B12, and antiepileptics [37]. 

## 4. Discussion

This study investigated the potential interplay between FA and the SARS-CoV-2 viral infection. We used studies based on molecular docking, in silico models, and real-world studies. Data from in silico studies and molecular docking supported that FA inhibits SARS-CoV-2 entry into the host and viral replication, binding at essential residues. On the other hand, real-world data showed contradictory results. A protective role of FA supplementation against SARS-CoV-2 infection has been indicated. However, observational studies indicate that dispensed FA supplementations due to deficits during systemic inflammation from SARS-CoV-2 increase the risk of mortality after the infection.

Studies based on in silico models have the advantage that they can lead to fast predictions concerning a large set of compounds in a high-throughput mode. At the same time, they combine the advantages of both in vivo and in vitro experimentation without subjecting themselves to ethical considerations and lack of control associated with in vivo experiments [38]. In addition, in silico models allow scientists to include a virtually unlimited parameter, rendering the outcomes more applicable to the organism, and they mainly are used in pharmacokinetic experimentation [38]. 

To the contrary, this kind of model has certain disadvantages. In in silico models and molecular docking studies, the most significant limitation is associated with the lack of confidence in scoring functions’ ability [39].

According to patients’ studies [35,36], it is possible that both inadequate and/or excessive amounts of folate may be detrimental to host resistance to SARS-CoV-2 infection and that there may be an optimal range of physiological folate status related to host resistance to COVID-19 infection and severity. Data support that folate supplementation increases viral replication in host cells and increases infection, hospitalization, and mortality. However, folate deficiency may also be harmful due to its effects on immune function and other important physiological pathways that protect against SARS-CoV-2 infection and progression [35,36]. Therefore, a major conundrum must be resolved regarding the optimal range of folate levels that are protective against SARS-CoV-2 infection, hospitalization, and mortality. Furthermore, according to Acosta-Elias et al., it is also critical to account for race/ethnicity, given the increased risk for infection, hospitalization, and mortality among black women compared to white women [35]. 

Although impressive progress has been achieved in the last years within the folate pathway, many questions remain to be answered. Why can the normal concentration of FA not inhibit the entry of SARS-CoV-2 into target cells? Additionally, what concentrations of FA would have to be given above what is already in the blood to be effective? Folate is especially and mainly carried in the blood in red blood cells. It has multiple mechanisms of uptake into cells that complicate thinking about what blood concentrations would be required to inhibit SARS-CoV-2 entry into target cells effectively. In that context, we should also take into account that a high intake of FA may have adverse effects, such as masking a vitamin B12 deficiency. High ingestion has also been related to carcinogenesis or cancer recurrence in certain groups [13].

There is also emerging knowledge that FA supplements impact other viral infections, including the human immunodeficiency virus (HIV), the hepatitis B virus, and the Zika virus infection, which are all included [40,41,42].

In that context, it has been elucidated that FA deficiency has been associated with HPV-associated atypical squamous cells of undetermined significance [43]. HPV infection persists and progresses to cervical dysplasia in females, given that folate deficiency is linked to insufficient cellular immunity [38]. Moreover, folate receptor alpha (FRalpha) has been described as a factor involved in mediating Ebola virus entry into cells [43], resulting in internalization and subsequent viral ingress into the cytoplasm via caveolae [44].

Furthermore, Vilaceka et al. hypothesized that low folate values are associated with hyperhomocysteinemia related to the risk of a sudden stroke in a group of HIV-infected children, as FA is required for the remethylation of homocysteine to methionine [45]. Besides, folate plays a key role in DNA synthesis, repair, and methylation, forming the basis of mechanistic explanations for the putative role of folate in cancer prevention [46]. Since the discovery of sulfa drugs and trimethoprim, the folate pathway has been a critical target for developing new drugs against infectious diseases [47]. Recently, it has been suggested that folate-conjugated HSV G207 presents a folate receptor-targeted oncolytic virus with an increased anti-tumor efficiency and tumor targeting specificity compared to the naked HSV, with a potential therapeutic value via retargeting to tumor cells [48,49,50]. Authors should discuss the results and how they can be interpreted from the perspective of previous studies and the working hypotheses. The findings and their implications should be discussed in the broadest context possible. Future research directions may also be highlighted.

A major limitation of our study is associated with the fact that the number of patient studies is quite a few. In addition, the studies included in our review article are mainly observational studies, which are prone to bias and cannot show causality. As a result, even though their outcomes might provide us with valuable medical data, more extensive studies are imperative to validate the demonstrated findings. Moreover, it is essential to raise the awareness of scientists and physicians regarding FA and its potential impact on SARS-CoV-2 infection. One aspect that could provide significant information could be the profound investigation of FA’s role in conditions such as the long COVID period and/or variants.

## 5. Conclusions

Generally, data support that vitamin deficiency increases host susceptibility to viral infections. In this study, data deriving from in silico studies and molecular docking support that FA could be considered a potential inhibitor for the SARS-CoV-2 virus’ entry into host cells and viral replication, binding at essential residues. Accordingly, in patients’ studies, a protective role of FA supplementation in case of deficiency against SARS-CoV-2 infection has been indicated. However, there are also contradictory data from observational studies indicating that dispensed FA supplementations might increase the risk of mortality. This information is related to processing the hypothesis that inadequate and/or excessive amounts of folate may be detrimental to host resistance to SARS-CoV-2 infection and that there may be an optimal range of physiological folate status related to host resistance to COVID-19 infection and severity. Hence, more questions remain to be answered about the intriguing interplay between FA and SARS-CoV-2 viral infection, as well as primarily what are the normal concentrations of FA that cannot inhibit the entry of SARS-CoV-2 into target cells, as well as what concentrations of FA would have to be given above what is already in the blood to be effective. It would be quite interesting if future randomized controlled clinical trial studies could reveal the potential association between FA and SARS-CoV-2 infection, antibody titers, as well as the effect of FA in infected but fully or partially vaccinated patients against SARS-CoV-2. In addition, it would be of great importance to investigate if other forms of folate could impact SARS-CoV-2 infected subjects and their antibodies or health outcomes and if FA could have a potential and beneficial impact on the existing treatment armamentarium against SARS-CoV-2 viral infection. It seems that this intriguing interplay between these two entities is going to concern both laboratory scientists and everyday clinical physicians who are occupied with this specific viral infection.

## Figures and Tables

**Figure 1 jpm-13-00561-f001:**
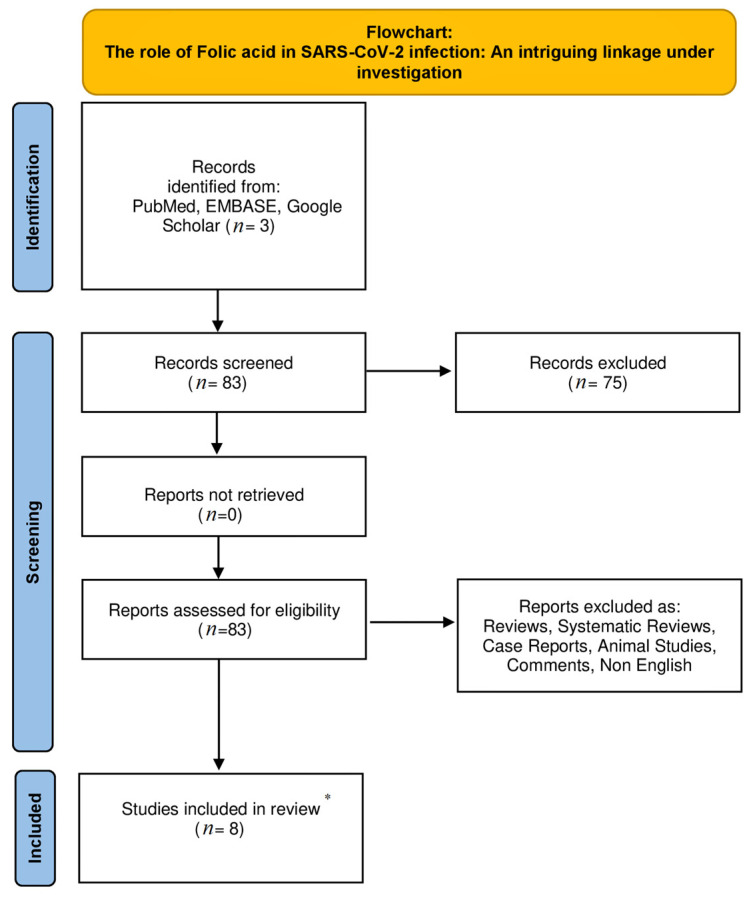
The flowchart of the study. * Only original non-animal studies written in English were included.

**Table 1 jpm-13-00561-t001:** Studies in silico and molecular docking concerning the potential association between folic acid and SARS-CoV-2.

Author/[Ref]	Year of Study	Study Design	Main Outcomes
Chen et al./[30]	2022	Target drugs, hub genes, molecular docking	FA was potentially an antagonist of SARS-CoV-2 N, but its effect on viruses is unclear
Ugurel et al./[31]	2020	Genome sequences, protein–drug interactions, in silico methods	FA was among the most promising drugs, having the potency to inhibit both the wild type and mutant SARS-CoV-2 helicase
Serseg et al./[32]	2021	Molecular docking	Hispidin, lepidine E, and FA inhibited the main protease of the virus
Kumar et al./[33]	2021	Molecular docking	FA alone, or in combination with its derivates, such as tetrahydrofolic acid and 5-methyl tetrahydrofolic acid, were potential molecules against COVID-19 infection
Eskandari/[34]	2022	Molecular docking, in silico methods	FA, among other vitamins, inhibited SARS-CoV-2 entry into the host and viral replication, binding at important residues

Abbreviations: COVID-19, coronavirus disease 2019; FA: folic acid.

**Table 2 jpm-13-00561-t002:** Patients’ studies concerning the potential interplay between folic acid and SARS-CoV-2 viral infection.

Author/[Ref]	Year	Study Design	Study Population	Main Findings
Acosta-Elias et al./[35]	2020	Comparative study	Pregnant vs non pregnant women suffering from influenza vs COVID-19	Protective role of FA against SARS-CoV-2 infection
Topless et al./[36]	2022	Case control analysis	Data from 380.380 UKBB participants with general practice prescription data for 2019–2021	Relation between COVID-19 diagnosis, COVID-19-related death, and FA oral prescription. Methotrexate diminished the relation
Bliek-Buen et al./[37]	2021	Retrospective, observational study	An amount of 8570 subjects, from two regions in Spain and Italy	Oral FA supplementationwas significantly associated with 30-day mortality in subjects with COVID-19 in both regions

Abbreviations: COVID-19, coronavirus disease 2019; FA: folic acid; UKBB: UK Biobank.

## Data Availability

Not applicable.

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
