# Peer review of "The Role of Folic Acid in SARS-CoV-2 Infection: An Intriguing Linkage under Investigation"

_jpm, 2023, doi:10.3390/jpm13030561_

Round 1

Reviewer 1 Report

Review MDPI Manuscript 2227765

This is an interesting non-systematic review examining the existing literature (not exhaustive) about supplementation with folate and the risk of SARS-CoV-2 infection, hospitalization and mortality. The authors point out the interesting observation that in silico and in vitro studies suggest that folate (and related derivatives) may have competitive and other type of inhibitory binding activity on several of the proteins expressed by the SARS-CoV-2 genome; however, large scale epidemiological studies suggest that there is increased risk with folate supplementation (Topeless et al , 2019, BMJ Open) for infection, hospitalization and mortality. This is in contrast to other studies that demonstrate that pregnant women were less likely to need in-patient care or hospitalization due to SARS-CoV-2 infection with the suggestion that folate supplementation might be at the root of this observed protection from hospitalization compared to non-pregnant women. Because of the many intrinsic roles folate plays in myriad metabolic pathways and one-carbon transfers in a variety of physiological systems, cells types and tissues it is not surprising that its role in SARS-CoV-2 infection, hospitalization and mortality is complex and is affected by many factors.  

Below are my comments on this manuscript which I believe is of value but needs some tightening up and some minor revisions.

There are many word usage errors that must also be fixed so this must be gone over by a native English speaking editor:

For example:

Line 31: What does the term “widowing” mean

            Line 42: Wrong word use “antigenic,” perhaps you mean “antigen?”

            Line 53: Replace “due to their ability” to “due to its ability”

Line 113: Replace “is known for involving in counteracting” with “is known for its involvement in counteracting”

Line 137: Replace “In thaw context” with “In this context

Line 76: In the Materials and Methods Section, it would be better to have some more detail. How many articles were retrieved using the key word searches in total, how many were rejected for what reasons, for example animal studies. Even though this is a non-systematic review, a flow chart should be included of the key word searches, what databases were searched, how many total articles were identified, which one were rejected and for what reason, and the final number of articles examined. This allows other interested researchers to conduct their own searchers or try to repeat what you have done in order to confirm and extend what has been done here. It is also important to state the exact queries used and the Boolean operators and exactly which combinations, such as the following: “SARS-CoV-2”  OR “COVID-19” AND “folic acid” OR “folate” OR “vitamin B9.” How many articles were retrieved for the period 2020 to 2022 with each combination?

In considering neutraceuticals as potential therapeutics for SARS-COV-2, including folic acid, there is an apparent conundrum that must  be dealt with because folic acid is an essential nutrient already circulating in the blood and body fluids and in all human cells as part of its normal function in molecular biosynthesis of amino acids and deoxyribonucleic acid and other metabolic pathways as a key methyl donor, so de facto we must be talking about giving large doses of folic acid to prevent infection. Therefore, what is the normal concentration of folic acid in blood and body fluids and what is its normal intracellular concentration? And why is the normal concentration of folic acid not already inhibiting entry of SARS-COV-2 into target cells? What is the half-life of folic acid in blood and extracellular fluid based on dosing studies and what is considered deficiency of folic acid in terms of blood and extracellular fluids?

As found at the Mount Sinai Medical System website (https://www.mountsinai.org/health-library/tests/folic-acid-test):

“The normal range is 2.7 to 17.0 nanograms per milliliter (ng/mL) or 6.12 to 38.52 nanomoles per liter (nmol/L).           Normal value ranges may vary slightly among different labs. Talk to your provider about the meaning of your test results.The examples above show the common measurements for results for these tests. Some labs use different measurements or may test different specimens.”

So what were the binding energies for the various targets of SARS-CoV-2 observed in the neutraceutical and molecule screening and docking studies and how did they relate to normal circulating levels of folic acid. Seeing that a normal range of blood folic acid is 270 to 1700 nanograms per 100mLs of blood, it is critical to understand the IC50s of these molecular docking studies to give some sense of what concentrations of folic acid would have to be given above what is already in the blood to be effective. Also the fact that folate as opposed to cobalamin is mainly carried in the blood in red blood cells and has multiple mechanisms of uptake into cells complicates thinking about what blood concentrations would be required to effectively inhibit SARS-COV-2 entry into target cells (https://www.ncbi.nlm.nih.gov/pmc/articles/PMC4867132/).

Paragraph 151 to 161: In the Acosta-Elias study it is important to not overstate the difference between pregnant women and non-pregnant women getting hospitalized for COVID-19; in the UK data the group analyzed the ratio was 0.9177 (pregnant)/0.9659 (non-pregnant) = 0.95—this is not a huge difference. It is also why the Spanish authors keep making the comparison between the current SARS-CoV-2 pandemic and the H1N1 influenza pandemic and the associated 10-fold difference between pregnant versus non-pregnant women requiring inpatient care (pregnant women are 10 gold less likely to need in patient care during COVID-19 than H1N1 compared to non-pregnant women. It would seem to this reviewer that the most important comparison is the current SARS-CoV-2 between pregnant versus non-pregnant women. That paper would have been much more powerful if they had actually measured folic acid in blood. It was also of great intrigue and interest that pregnant black women in the same UK study were 8-fold more likely to get to require inpatient care than their white pregnant women counterparts, and it was pointed out the genetic differences may account for differences in folate metabolism, absorption and Red Blood Cell (RBC) concentrations, which means that any clinical trial conducted would certainly have to account for differences in race/ethnicity where it would be predicted from the UK study that pregnant black women may require higher dosing of folate to achieve the same level of protection from SARS-CoV-2, or that availability and cost of folate supplementation may prevent use in pregnant women of color in underserved and marginalize communities.

Line 159-161: It is unclear why the authors state the Acosta-Elias study suggests that Folate might inhibit replication of the virus as opposed to blocking entry by occupying sites on the spike protein and interfering with the receptor binding domain, or the many other in silico examples of potential targets of Folate on various proteins expressed by the SARS-CoV-2 virus. However, as stated by Topeless et al. (BMJ Open, 2022), it is highly possible that folate supplementation aids production of large amounts of virus through a simple biosynthetic role, thus calling into question all the in silico findings mentioned above. The fact that methotrexate (an anti-folate) reverses the effects of folate supplementation strongly supports this notion that folic acid is promoting the production of increased viral replication inside cells and it not surprising.

“With the caveat that our study is observational epidemiology and causality cannot be inferred, our study does support the possibility that external folate supply facilitates the production of large amounts of virus, contributing to clinical infection and mortality. With the same caveat on inference of causality from observational data, our study also supports the notion that SARS-CoV-2 replication is enhanced by folate supply based on our finding that coprescription of an antifolate (methotrexate) attenuated the association of supplementation with folic acid with COVID-19 outcomes.”

This also suggest that the studies demonstrating lower hospitalization rates and in-patient care in pregnant versus non-pregnant women must be due to some other factor of pregnancy, perhaps hormonal, that is as yet unidentified.

As discussed by these authors and by Topeless et al. (BMJ Open, 2022), there may be an optimum range of folate to provide protection against SARS-CoV-2 infection, where too much folate supplementation is deleterious and folate insufficiency is also deleterious due to effects on immune function:

“It is therefore possible that both inadequate and excessive amounts of folate may be detrimental to host resistance to SARS-CoV-2 infection and that there may be an optimal range of physiological folate status related to host resistance to COVID-19 infection and severity.”

Ultimately the authors are correct that there is a major conundrum here that must be resolved perhaps with larger and more independent epidemiological studies to confirm that folate supplementation increases viral replication in host cells and increases infection, hospitalization and mortality; but also that folate deficiency may also be harmful due to effects on immune function and other important physiological pathways that protect against SARS-C0V-2 infection and progression.

It would appear that the most impactful study would be epidemiological studies to measure blood folate levels across thousands of individuals and to examine and follow their outcomes in terms of SARS-CoV-2 infection, hospitalization and mortality in order to confirm whether, as suggested by Topeless et al (BMJ Open, 2019), there is indeed an optimal range of folate levels that are protective against SARS-CoV-2 infection, hospitalization and mortality, and being outside that range (below or above) promotes infection, hospitalization and increased mortality. As pointed out by the Acosta-Elias study, it is also critical to account for race/ethnicity given the dramatically increased risk for infection, hospitalization and mortality found among black women compared to white women. This could be easily included data in any standard epidemiological study.

Line 238: This needs to be restated from, “FA interferes with the remethylation of homocysteine to methionine” to for example, “FA is required for the remethylation of homocysteine to methionine.”

In references for example the recent Zhang article (2022) does not appear to be included suggesting additional mechanisms of folate inhibition of SARS-CoV-2 or promotion of SARS-CoV-2 infection and progression:

Zhang Y, Pang Y, Xu B, Chen X, Liang S, Hu J, Luo X. Folic acid restricts SARS-CoV-2 invasion by methylating ACE2. Front Microbiol. 2022 Aug 17;13:980903. doi: 10.3389/fmicb.2022.980903. PMID: 36060767; PMCID: PMC9432853.

Zhang Y, Guo R, Kim SH, Shah H, Zhang S, Liang JH, Fang Y, Gentili M, Leary CNO, Elledge SJ, Hung DT, Mootha VK, Gewurz BE. SARS-CoV-2 hijacks folate and one-carbon metabolism for viral replication. Nat Commun. 2021 Mar 15;12(1):1676. doi: 10.1038/s41467-021-21903-z. PMID: 33723254; PMCID: PMC7960988.

“Here, we show that SARS-CoV-2 remodels host folate and one-carbon metabolism at the post-transcriptional level to support de novo purine synthesis, bypassing viral shutoff of host translation. Intracellular glucose and folate are depleted in SARS-CoV-2-infected cells, and viral replication is exquisitely sensitive to inhibitors of folate and one-carbon metabolism, notably methotrexate. Host metabolism targeted therapy could add to the armamentarium against future coronavirus outbreaks.”

Lokhande KB, Doiphode S, Vyas R, Swamy KV. Molecular docking and simulation studies on SARS-CoV-2 Mpro reveals Mitoxantrone, Leucovorin, Birinapant, and Dynasore as potent drugs against COVID-19. J Biomol Struct Dyn. 2021 Nov;39(18):7294-7305. doi: 10.1080/07391102.2020.1805019. Epub 2020 Aug 20. PMID: 32815481; PMCID:

Perła-Kaján J, Jakubowski H. COVID-19 and One-Carbon Metabolism. Int J Mol Sci. 2022 Apr 10;23(8):4181. doi: 10.3390/ijms23084181. PMID: 35456998; PMCID: PMC9026976.

McCaddon A, Regland B. COVID-19: A methyl-group assault? Med Hypotheses. 2021 Apr;149:110543. doi: 10.1016/j.mehy.2021.110543. Epub 2021 Feb 18. PMID: 33657459; PMCID: PMC7890339.

Author Response

Response to Reviewer 1 comments:

  1. This is an interesting non-systematic review examining the existing literature (not exhaustive) about supplementation with folate and the risk of SARS-CoV-2 infection, hospitalization and mortality. The authors point out the interesting observation that in silico and in vitro studies suggest that folate (and related derivatives) may have competitive and other type of inhibitory binding activity on several of the proteins expressed by the SARS-CoV-2 genome; however, large scale epidemiological studies suggest that there is increased risk with folate supplementation (Topeless et al , 2019, BMJ Open) for infection, hospitalization and mortality. This is in contrast to other studies that demonstrate that pregnant women were less likely to need in-patient care or hospitalization due to SARS-CoV-2 infection with the suggestion that folate supplementation might be at the root of this observed protection from hospitalization compared to non-pregnant women. Because of the many intrinsic roles folate plays in myriad metabolic pathways and one-carbon transfers in a variety of physiological systems, cells types and tissues it is not surprising that its role in SARS-CoV-2 infection, hospitalization and mortality is complex and is affected by many factors.

Response: We sincerely thank you for your kind words about our paper. We are delighted to receive a positive feedback from you.

  1. Below are my comments on this manuscript which I believe is of value but needs some tightening up and some minor revisions. There are many word usage errors that must also be fixed so this must be gone over by a native English speaking editor:

Response: We have gone through the whole manuscript and all grammatical mistakes and misspellings were corrected. Signs of misunderstanding were also rephrased. We hope that you find these revisions an improvement.

  1. For example: Line 31: What does the term “widowing” mean

Response: This word has been changed from widowing to targeted.

  1. Line 42: Wrong word use “antigenic,” perhaps you mean “antigen?”

Response: Thank you for this remark. It has been corrected.

  1. Line 53: Replace “due to their ability” to “due to its ability”

Response: Thank you for this point. It has been corrected.

  1. Line 113: Replace “is known for involving in counteracting” with “is known for its involvement in counteracting”

Response: It has been corrected, thank you.

  1. Line 137: Replace “In thaw context” with “In this context

Response: Thank you, in our revised manuscript it has been corrected.

  1. Line 76: In the Materials and Methods Section, it would be better to have some more detail. How many articles were retrieved using the key word searches in total, how many were rejected for what reasons, for example animal studies. Even though this is a non-systematic review, a flow chart should be included of the key word searches, what databases were searched, how many total articles were identified, which one were rejected and for what reason, and the final number of articles examined. This allows other interested researchers to conduct their own searchers or try to repeat what you have done in order to confirm and extend what has been done here. It is also important to state the exact queries used and the Boolean operators and exactly which combinations, such as the following: “SARS-CoV-2” OR “COVID-19” AND “folic acid” OR “folate” OR “vitamin B9.” How many articles were retrieved for the period 2020 to 2022 with each combination?

Response: Thank you for this comment. A flowchart has been added in revised manuscript, as suggested.

  1. In considering neutraceuticals as potential therapeutics for SARS-COV-2, including folic acid, there is an apparent conundrum that must be dealt with because folic acid is an essential nutrient already circulating in the blood and body fluids and in all human cells as part of its normal function in molecular biosynthesis of amino acids and deoxyribonucleic acid and other metabolic pathways as a key methyl donor, so de facto we must be talking about giving large doses of folic acid to prevent infection. Therefore, what is the normal concentration of folic acid in blood and body fluids and what is its normal intracellular concentration? And why is the normal concentration of folic acid not already inhibiting entry of SARS-COV-2 into target cells? What is the half-life of folic acid in blood and extracellular fluid based on dosing studies and what is considered deficiency of folic acid in terms of blood and extracellular fluids? As found at the Mount Sinai Medical System website (https://www.mountsinai.org/health-library/tests/folic-acid-test): “The normal range is 2.7 to 17.0 nanograms per milliliter (ng/mL) or 6.12 to 38.52 nanomoles per liter (nmol/L). Normal value ranges may vary slightly among different labs. Talk to your provider about the meaning of your test results. The examples above show the common measurements for results for these tests. Some labs use different measurements or may test different specimens.”

Response: We greatly appreciate your comments and constructive suggestions which help us improve the quality of the manuscript.

  1. So what were the binding energies for the various targets of SARS-CoV-2 observed in the neutraceutical and molecule screening and docking studies and how did they relate to normal circulating levels of folic acid. Seeing that a normal range of blood folic acid is 270 to 1700 nanograms per 100mLs of blood, it is critical to understand the IC50s of these molecular docking studies to give some sense of what concentrations of folic acid would have to be given above what is already in the blood to be effective. Also the fact that folate as opposed to cobalamin is mainly carried in the blood in red blood cells and has multiple mechanisms of uptake into cells complicates thinking about what blood concentrations would be required to effectively inhibit SARS-COV-2 entry into target cells (https://www.ncbi.nlm.nih.gov/pmc/articles/PMC4867132/).

Response: Thank you for all the issues you arise, which help us improve the quality of the manuscript. In the revised manuscript, we have included all these questions as food for thought in the discussion section.

  1. Paragraph 151 to 161: In the Acosta-Elias study it is important to not overstate the difference between pregnant women and non-pregnant women getting hospitalized for COVID-19; in the UK data the group analyzed the ratio was 0.9177 (pregnant)/0.9659 (non-pregnant) = 0.95—this is not a huge difference. It is also why the Spanish authors keep making the comparison between the current SARS-CoV-2 pandemic and the H1N1 influenza pandemic and the associated 10-fold difference between pregnant versus non-pregnant women requiring inpatient care (pregnant women are 10 gold less likely to need in patient care during COVID-19 than H1N1 compared to non-pregnant women. It would seem to this reviewer that the most important comparison is the current SARS-CoV-2 between pregnant versus non-pregnant women. That paper would have been much more powerful if they had actually measured folic acid in blood. It was also of great intrigue and interest that pregnant black women in the same UK study were 8-fold more likely to get to require inpatient care than their white pregnant women counterparts, and it was pointed out the genetic differences may account for differences in folate metabolism, absorption and Red Blood Cell (RBC) concentrations, which means that any clinical trial conducted would certainly have to account for differences in race/ethnicity where it would be predicted from the UK study that pregnant black women may require higher dosing of folate to achieve the same level of protection from SARS-CoV-2, or that availability and cost of folate supplementation may prevent use in pregnant women of color in underserved and marginalize communities.

Response: Thank you for the great comments that have been included in the revised discussion section.

  1. Line 159-161: It is unclear why the authors state the Acosta-Elias study suggests that Folate might inhibit replication of the virus as opposed to blocking entry by occupying sites on the spike protein and interfering with the receptor binding domain, or the many other in silico examples of potential targets of Folate on various proteins expressed by the SARS-CoV-2 virus.

Response: Thank you for this point. We have deleted this unclear point in the revise manuscript.

  1. However, as stated by Topeless et al. (BMJ Open, 2022), it is highly possible that folate supplementation aids production of large amounts of virus through a simple biosynthetic role, thus calling into question all the in silico findings mentioned above. The fact that methotrexate (an anti-folate) reverses the effects of folate supplementation strongly supports this notion that folic acid is promoting the production of increased viral replication inside cells and it not surprising. “With the caveat that our study is observational epidemiology and causality cannot be inferred, our study does support the possibility that external folate supply facilitates the production of large amounts of virus, contributing to clinical infection and mortality. With the same caveat on inference of causality from observational data, our study also supports the notion that SARS-CoV-2 replication is enhanced by folate supply based on our finding that coprescription of an antifolate (methotrexate) attenuated the association of supplementation with folic acid with COVID-19 outcomes.” This also suggest that the studies demonstrating lower hospitalization rates and in-patient care in pregnant versus non-pregnant women must be due to some other factor of pregnancy, perhaps hormonal, that is as yet unidentified.

Response: We appreciate all these useful comments that have been incorporated in the revised manuscript.

  1. As discussed by these authors and by Topeless et al. (BMJ Open, 2022), there may be an optimum range of folate to provide protection against SARS-CoV-2 infection, where too much folate supplementation is deleterious and folate insufficiency is also deleterious due to effects on immune function: “It is therefore possible that both inadequate and excessive amounts of folate may be detrimental to host resistance to SARS-CoV-2 infection and that there may be an optimal range of physiological folate status related to host resistance to COVID-19 infection and severity.” Ultimately the authors are correct that there is a major conundrum here that must be resolved perhaps with larger and more independent epidemiological studies to confirm that folate supplementation increases viral replication in host cells and increases infection, hospitalization and mortality; but also that folate deficiency may also be harmful due to effects on immune function and other important physiological pathways that protect against SARS-C0V-2 infection and progression. It would appear that the most impactful study would be epidemiological studies to measure blood folate levels across thousands of individuals and to examine and follow their outcomes in terms of SARS-CoV-2 infection, hospitalization and mortality in order to confirm whether, as suggested by Topeless et al (BMJ Open, 2019), there is indeed an optimal range of folate levels that are protective against SARS-CoV-2 infection, hospitalization and mortality, and being outside that range (below or above) promotes infection, hospitalization and increased mortality. As pointed out by the Acosta-Elias study, it is also critical to account for race/ethnicity given the dramatically increased risk for infection, hospitalization and mortality found among black women compared to white women. This could be easily included data in any standard epidemiological study.

Response: Thank you for this direction. You raise a very valid point that is discussed in the revised manuscript, as suggested.

-------------------------------------------------------------------------------------------------------

  1. Line 238: This needs to be restated from, “FA interferes with the remethylation of homocysteine to methionine” to for example, “FA is required for the remethylation of homocysteine to methionine.”

Response: Thank you for your comment. The correction has been made.

  1. In references for example the recent Zhang article (2022) does not appear to be included suggesting additional mechanisms of folate inhibition of SARS-CoV-2 or promotion of SARS-CoV-2 infection and progression: Zhang Y, Pang Y, Xu B, Chen X, Liang S, Hu J, Luo X. Folic acid restricts SARS-CoV-2 invasion by methylating ACE2. Front Microbiol. 2022 Aug 17;13:980903. doi: 10.3389/fmicb.2022.980903. PMID: 36060767; PMCID: PMC9432853. Zhang Y, Guo R, Kim SH, Shah H, Zhang S, Liang JH, Fang Y, Gentili M, Leary CNO, Elledge SJ, Hung DT, Mootha VK, Gewurz BE. SARS-CoV-2 hijacks folate and one-carbon metabolism for viral replication. Nat Commun. 2021 Mar 15;12(1):1676. doi: 10.1038/s41467-021-21903-z. PMID: 33723254; PMCID: PMC7960988.

Response: Thank you for your comment. These articles are associated with animal models that are not included in the review.

  1. “Here, we show that SARS-CoV-2 remodels host folate and one-carbon metabolism at the post-transcriptional level to support de novo purine synthesis, bypassing viral shutoff of host translation. Intracellular glucose and folate are depleted in SARS-CoV-2-infected cells, and viral replication is exquisitely sensitive to inhibitors of folate and one-carbon metabolism, notably methotrexate. Host metabolism targeted therapy could add to the armamentarium against future coronavirus outbreaks.”

Lokhande KB, Doiphode S, Vyas R, Swamy KV. Molecular docking and simulation studies on SARS-CoV-2 Mpro reveals Mitoxantrone, Leucovorin, Birinapant, and Dynasore as potent drugs against COVID-19. J Biomol Struct Dyn. 2021 Nov;39(18):7294-7305. doi: 10.1080/07391102.2020.1805019. Epub 2020 Aug 20. PMID: 32815481; Perła-Kaján J, Jakubowski H. COVID-19 and One-Carbon Metabolism. Int J Mol Sci. 2022 Apr 10;23(8):4181. doi: 10.3390/ijms23084181. PMID: 35456998; PMCID: PMC9026976. McCaddon A, Regland B. COVID-19: A methyl-group assault? Med Hypotheses. 2021 Apr;149:110543. doi: 10.1016/j.mehy.2021.110543. Epub 2021 Feb 18. PMID: 33657459; PMCID: PMC7890339.

Response: We appreciate you taking the time to offer us your insights related to the paper. We found your feedback very constructive. We tried to be responsive to your concerns.

Reviewer 2 Report

Authors wrote an interesting and well structured paper. I find it suitable of publication in this important journal after minor revisions.

1. Introduction: update data on SAR CoV2 wordwilde at the day of resubmission

In addition, other experience describe the role of vitamin in infectious diseases outcome (see and cite Patti G, Pellegrino C, Ricciardi A, et al. Potential Role of Vitamins A, B, C, D and E in TB Treatment and Prevention: A Narrative Review. Antibiotics (Basel). 2021;10(11):1354. Published 2021 Nov 5. doi:10.3390/antibiotics10111354 and Papagni, Roberta et al. “Impact of Vitamin D in Prophylaxis and Treatment in Tuberculosis Patients.” International journal of molecular sciences vol. 23,7 3860. 31 Mar. 2022, doi:10.3390/ijms23073860)

Methods: please add the periodo of study (ex from March 2020 to 10 October 2022)

Results. clear in all part

Discussion: add some consideration on variant and long covid on SARS CoV2 outcome 

Conclusion: give some proposal that came from your interesting paper 

Author Response

Response to Reviewer 2 comments

  1. Authors wrote an interesting and well-structured paper. I find it suitable of publication in this important journal after minor revisions.

Response: We sincerely thank you for your kind words about our paper. We are delighted to receive positive feedback from you.

  1. Introduction: update data on SAR CoV2 wordwilde at the day of resubmission

Response: Thank you for this comment. We included update data on SAR CoV2 worldwide in the revised manuscript.

  1. In addition, other experience describes the role of vitamin in infectious diseases outcome (see and cite Patti G, Pellegrino C, Ricciardi A, et al. Potential Role of Vitamins A, B, C, D and E in TB Treatment and Prevention: A Narrative Review. Antibiotics (Basel). 2021;10(11):1354. Published 2021 Nov 5. doi:10.3390/antibiotics10111354 and Papagni, Roberta et al. “Impact of Vitamin D in Prophylaxis and Treatment in Tuberculosis Patients.” International journal of molecular sciences 23,7 3860. 31 Mar. 2022, doi:10.3390/ijms23073860)

Response: Thank you for these remarks. The appropriate additions have been made.

  1. Methods: please add the period of study (ex from March 2020 to 10 October 2022)

Response: Thank you for your comment. The period of the study has been added.

  1. Results: clear in all part

Response: Thank you for your positive comment.

  1. Discussion: add some consideration on variant and long covid on SARS CoV2 outcome 

Response: Thank you for this comment. We have added some consideration concerning these two aspects of SARS-CoV-2 infection.

  1. Conclusion: give some proposal that came from your interesting paper 

Response: Thank you very much for this suggestion. A proposal concerning the potential therapeutic profile of folic acid has been added.

We found your feedback very constructive. We tried to be responsive to your concerns. We really thank you for taking the time and energy to help us improve this paper.

Reviewer 3 Report

Thank you for the opportunity to review this manuscript. It is an exciting review. I have only some suggestions to make it better. The introuduction could be implemented with other information about using other vitamins in the SARS-COV-2. Here some recent reference concerning this topic."Sinopoli A, Caminada S, Isonne C, Santoro MM, Baccolini V. What Are the Effects of Vitamin A Oral Supplementation in the Prevention and Management of Viral Infections? A Systematic Review of Randomized Clinical Trials. Nutrients. 2022 Oct 1;14(19):4081. doi: 10.3390/nu14194081. PMID: 36235733; PMCID: PMC9572963", Bae, M., & Kim, H. (2020). The role of vitamin C, vitamin D, and selenium in immune system against COVID-19. Molecules25(22), 5346.".

In the methodology it's necessary to explicit the route of administration of folic acid.

Author Response

Response to Reviewer 3 comments

  1. Thank you for the opportunity to review this manuscript. It is an exciting review.

Response: We sincerely thank you for your kind words about our paper. We are delighted to receive positive feedback from you.

  1. I have only some suggestions to make it better. The introduction could be implemented with other information about using other vitamins in the SARS-COV-2. Here some recent reference concerning this topic."Sinopoli A, Caminada S, Isonne C, Santoro MM, Baccolini V. What Are the Effects of Vitamin A Oral Supplementation in the Prevention and Management of Viral Infections? A Systematic Review of Randomized Clinical Trials. Nutrients. 2022 Oct 1;14(19):4081. doi: 10.3390/nu14194081. PMID: 36235733; PMCID: PMC9572963", Bae, M., & Kim, H. (2020). The role of vitamin C, vitamin D, and selenium in immune system against COVID-19.Molecules25(22), 5346.".

Response: Thank you for this point. We have provided information about using other vitamins in sars-cov-2, in the revised introduction as suggested.

In the methodology it's necessary to explicit the route of administration of folic acid.

Response: Thank you for this remark. We have defined this point in the revised manuscript.

We found your feedback very constructive. We tried to be responsive to your concerns. We really thank you for taking the time and energy to help us improve this paper.